# Visualization of Subcutaneous Blood Vessels Based on Hyperspectral Imaging and Three-Wavelength Index Images

**DOI:** 10.3390/s23218895

**Published:** 2023-11-01

**Authors:** Mohammed Hamza, Roman Skidanov, Vladimir Podlipnov

**Affiliations:** 1Department of Information Technology, Samara National Research University, Moskovskoye Shosse 34, 443086 Samara, Russia; mohmmed.mohee.1986@gmail.com (M.H.); podlipnovvv@ya.ru (V.P.); 2IPSI RAS—Branch of the FSRC “Crystallography and Photonics” RAS, Molodogvardeiskaya St. 151, 443001 Samara, Russia

**Keywords:** blood vessels under the skin, hyperspectrometer, spectral analysis, index images, hypercube

## Abstract

Blood vessel visualization technology allows nursing staff to transition from traditional palpation or touch to locate the subcutaneous blood vessels to visualized localization by providing a clear visual aid for performing various medical procedures accurately and efficiently involving blood vessels; this can further improve the first-attempt puncture success rate for nursing staff and reduce the pain of patients. We propose a novel technique for hyperspectral visualization of blood vessels in human skin. An experiment with six participants with different skin types, race, and nationality backgrounds is described. A mere separation of spectral layers for different skin types is shown to be insufficient. The use of three-wavelength indices in imaging has shown a significant improvement in the quality of results compared to using only two-wavelength indices. This improvement can be attributed to an increase in the contrast ratio, which can be as high as 25%. We propose and implement a technique for finding new index formulae based on an exhaustive search and a binary blood-vessel image obtained through an expert assessment. As a result of the search, a novel index formula was deduced, allowing high-contrast blood vessel images to be generated for any skin type.

## 1. Introduction

In recent years, hyperspectral imaging has been increasingly utilized in medicine. In biomedical applications [1,2], the noncontact, nondestructive diagnostic technology that relies on hyperspectral imaging is an emerging approach with a wide potential as a noninvasive diagnostic and prognostication technique. At definite wavelengths, biological tissues with different chemical content and physical characteristics have different reflectance and absorption, leading to different characteristic peaks and dips in their spectrum. An analysis of spectral signals provides an instrument for extracting qualitative and quantitative information about the tissue condition. Meanwhile, various tissue pathologies can be visualized by analyzing spatial distribution information provided by hyperspectral imaging, which diagnoses the tissue pathology. Hyperspectral microscopy enables the analysis of biological samples by acquiring spectral information at each pixel [3]. In medicine, hyperspectral visualization has been increasingly used for disease detection, in vitro diagnostics, surgical guidance, scientific research, and so on. There is also a non-medical use where obtaining a high-contrast image of hand blood vessels is important, namely, in-person identification methods [4,5,6,7]. Hyperspectral imaging can find the best contrast between the skin and blood vessels located 3–5 mm below the skin in the near-infrared (NIR) range [8].

Visualizing subcutaneous blood vessels can help diagnose and monitor various vascular diseases, such as deep vein thrombosis, varicose veins, and peripheral arterial disease, because human blood vessels contain medical information on the health condition or possible disease, making the development of an accurate visualization system and image processing algorithms highly relevant. Accurate vein detection ensures to insert the needle successfully and this particularly important in blood donation and kidney dialysis procedures. For nearly 80 percent of all those receiving in- and out-patient treatment, intravenous catheters need to be installed to draw or transfuse blood, deliver drugs, and infuse fluids [9,10]. In the study [11], the high failure rate of approximately 14 million cases on the first attempt out of 500 million annual insertions is concerning.

Overall, medical practitioners face several challenges during venipuncture for drug delivery. These challenges are influenced by factors that cause difficulties during this procedure, such as vein size, depth, patient age, skin color, obesity level, and dehydrated patients [12,13].

Moreover, unsuccessful needle insertions can lead to cuts on the bone, hematoma, bleeding, risk of infection, and increased pain and discomfort for the patients [14]. Additionally, with repeated failed attempts at needle insertion, patients and their families can become agitated, making the task even more arduous and adding to the cost of care, as well as potentially prolonging the procedure [15]; this can lead to a breakdown in the patient–nursing staff relationship, making it more difficult to establish trust and cooperation for future procedures. This delay can be particularly problematic in emergency situations where time is of the essence.

The enhancement of vein contrast relies on the fact that veins and surrounding tissue absorb and scatter differently NIR light. In the NIR range, light can easily penetrate the skin and subcutaneous fat due to low absorption, being absorbed or forward scattered by blood and uniformly scattered in all directions in subskin fat. Hence, in an image, the blood appears dark on the background of lighter skin and fat [16].

Indeed, many techniques have been developed that combine different methods for the visualizing subcutaneous blood vessels. [17,18,19,20,21,22,23,24,25,26]. In addition, these technologies can help healthcare professionals locate and access veins more accurately. However, they involve the use of expensive equipment, are complicated, not portable, require of the patients to stay motionless for quite a long time, and may require specialized training or the involvement of specialists to perform effectively the imaging procedures.

Additionally, there are handheld device technologies available that use NIR imaging for vein visualization [27,28]. It can be particularly useful in situations where traditional methods may not be accessible or effective. Dark skin tones (African American) and obesity can absorb more light, making it difficult for the device to detect veins [29]. Hair, scars on the skin, and tattoos can also interfere with the imaging process and make it difficult to visualize veins [30]. While AccuVein is effective at imaging superficial vessels, it may not provide as clear of a view for deeper veins [28]

The NIR imaging technique can be utilized to detect blood vessels, and an appropriate optical window for this purpose is typically selected within the range of 700 to 950 nm. NIR light has the advantage of being able to penetrate deeper into tissues compared to visible light, allowing for better visualization of blood vessels beneath the skin [31,32]. The trans-illuminations method involves shining a light through the skin to visualize veins. It is a simple and inexpensive technique but may not provide sufficient contrast for accurate vein identification, especially in patients with darker skin tones [33].

Note that there is an approach that utilizes a tailored narrow-band illumination instead of extracting spectral ranges of the hyperspectrometer [34]. Another reported technique for blood vessel imaging utilized the illumination of skin with a special high-power white source [35]. However, as the authors mentioned, the method is of limited use and only suitable for visualizing children’s skin because even high-power illumination is insufficient to penetrate an adult’s thick skin. Another concern stems from the fact that the method is deemed unsafe for all body parts except for limbs, with a “~200-W” power being concentrated on a relatively small body area with an optical fiber. The development of autonomous robotic venous access devices is still in its early stages, and further research and validation are needed before widespread adoption in the future [36,37,38].

Hyperspectral imaging for blood vessel detection was utilized for the first time as far back as 2009. The study described the visualization of the abdominal aorta and inferior vena cava in a spectral range of 650–700 nm; hyperspectral imaging in this context would involve capturing images at multiple narrow spectral bands within this range to obtain detailed spectral information about the blood vessels [39]. The monitoring of subskin blood vessels using hyperspectral imaging was utilized to determine blood oxygen levels in some parts of the human body [40,41,42,43]. To date, a multitude of works have been published reporting the use of NIR hyperspectral imaging for the delineation of skin blood vessels [44,45,46,47,48,49,50,51,52]. These works reported similar results but also revealed a number of common challenges. The major problem of this technique based on the simple separation of a spectral layer at one wavelength it works just for a particular skin type. In addition, this may limit the ability of hyperspectral imaging to image subcutaneous blood vessels in children.

In this work, we made an attempt to design a fairly universal technique for blood vessel visualization based on hyperspectral imaging using an index approach that has been widely employed for vegetation analysis [53,54,55,56,57,58,59,60,61]. A hyperspectral imaging index approach involves using hyperspectral imaging data for a more detailed analysis of the spectral characteristics of the scene or subject being imaged. In the context of blood vessel visualization, a hyperspectral imaging index approach can be employed by analyzing the spectral reflectance or absorption patterns across different wavelengths; indices can be derived to highlight blood vessels. Utilizing an index approach makes it possible to extract meaningful information from complex hyperspectral data and present it in a more interpretable form. In general, the process of creating index images involves applying mathematical operations to the pixel values of the input images. These operations can include subtracting or dividing pixel values, applying logarithmic or exponential functions, or using specific algorithms designed for the desired purpose and specific wavelength selection to enhance certain features or characteristics in the resulting image. There are quite complex methods based on neural networks [62]. In our current work, we sought to develop an extremely simple method that is fundamentally different from neural networks.

Previously, a method for directly generating index images has been proposed [63,64] and utilized for imaging skin blood vessels [64]. However, the method described in [64] also lacks universality because it described experiments conducted with the skin of a single person and had limitations relating to the number of wavelengths used (a two-wavelength index). Another study proved the possibility of using an index approach to detect skin regions in non-visible spectral imagery for search and rescue applications [65]. In this work, the study was conducted using the skin of six persons coming from three major races, and index images generated using three wavelengths were analyzed.

## 2. Materials and Methods

### 2.1. The Method of Obtaining Hyperspectral Images

In the context of this work, the scanning was carried out using a spectral scanning slit hyperspectral camera based on an Offner optical scheme (Samara National Research University., Samara, Russia) mounted on a rotary platform by installing a hyperspectral camera on a special filming tripod equipped with an angular rotation drive with the ability to set the rotation speed in the range of 2–3 rpm, that allows spatial scanning with a spectral resolution of 2.4 nm [66]. The Offner scheme is the most promising due to its simplicity, small dimensions, and high optical performance. With its help, it is possible to achieve a reduction in chromatic aberrations and distortion to a low level over a relatively wide spectral range while maintaining the compact dimensions of the hyperspectrometer. In the simplest case, the Offner spectrometer contains two concentric mirrors. A lens with a fixed focal length MIR-1V 2.8/37 (Vologodskiy Optiko-Mekhanicheskiy Zavod., Vologda, Russia) was selected, with an aperture value set to approximately 3.2 and a photosensitive matrix CMV4000 (Baumer GmbH., Friedberg, Germany), receiving images in both the visible and near-IR ranges.

The scanning time of a Hyperspectral Imaging Camera for one subject is about 40 s. Our targeted area was the forearm region of the subject; hence, around 30° of the scanning range (which covers the region from the elbow to the tip of the subject’s fingers) was selected. The fixed distance of the hyperspectral camera was set to be about 110 cm away from each participant’s forearm, with the forearm slanted at an angle of +30° and the camera tilted at an angle −30°. The lamp was installed at a distance of about 120 cm from the forearm, as close as possible to the optical axis of the experimental setup. Keeping the hand stable during the imaging process in front of a dark background is necessary to ensure accurate and reliable results when recording images from different spectral ranges. A hyperspectral image was obtained by recording the reflections or emissions of hundreds for different narrow bands; each pixel in a hyperspectral image contains a full spectrum of information.

The authors strived to achieve uniform illumination across the forearm being imaged. Careful calibration and adjustment of the illumination setup are essential to ensure consistent illumination intensity across the entire surface and hand Curvature Compensation. When 1000 W of halogen lamp was applied, the illumination of skin areas of interest was about 800 W/m2. This illumination approximately corresponds to natural illumination on a cloudless day.

The images were taken in 250 spectral channels, giving an average distance of 0.66 nm between each center wavelength. The maximum frame rate the camera can operate at is up to 10 frames per second, ranging from 400 to 1000 nm, with a spatial resolution of 1020 × 1022. Basler software (Basler., Ahrensburg, Germany) was used to configure camera parameters and collect data.

The captured data was then assembled into a three-dimensional data set that includes two spatial dimensions (image components) and one spectral dimension called a hyperspectral data cube (Figure 1).

Skin tone is an important factor that affects the vein localization process. Individuals with darker skin tones often have more melanin, a pigment that gives color to the skin; this can make it more difficult for healthcare professionals to visually locate veins, as darker skin may obscure the veins or make them less visible. In order to achieve better results of vein viewing in subjects with different skin tones, we classified the skin into four classes. The experiment included six male participants with different ages, racial backgrounds, and nationalities (see Table 1).

Conditions for hyperspectral imaging were identical for all the participants. In hyperspectral imaging, filming the same forearm multiple times for the same person may not be necessary or beneficial because hyperspectral imaging captures a wide range of spectral information for each pixel in an image. There is no practical need to film the same forearm multiple times. The obtained images would likely be almost identical, as the spectral properties of the forearm are not expected to change significantly within a short period of time. This approach saves time and increases the reliability and accuracy of the results in this study.

To calibrate the wavelength of light in our experiment, we utilized lasers with specific wavelengths of 532 nm and 633 nm. Additionally, we employed the Color Checker Video X-Rite for calibrating the spectral brightness coefficients of the channels in the final hypercube and the uniformity of illumination across the field of view [67]. Also, in our study, the RMS error evaluated the accuracy of image processing algorithms or models. It is calculated by taking the square root of the average of the squared differences between the actual and predicted values, with lower RMS error indicating better accuracy [64,68]. An experimental setup is indicated in Figure 2.

### 2.2. The Use of Spectral Indices

Usually, in hyperspectral visualization of various objects with characteristic spectral features, the method of selecting these objects is used based on index images. Spectral indices are algebraically expressed (sometimes as simple rational fractions; see Table 2) combinations of reflectance coefficients of the object of interest at two or more spectral regions that reflect relative values of target characteristics of the object under study [58]. Using a combination of brightness values in the selected spectral channels, an index image is generated corresponding to the index value in each pixel. To calculate the index, either simple differences, relations, or normalized difference relations can be used. The choice of channels can be determined on the basis of physical properties or empirically, which carry the necessary information for the selection of the object under study and are less susceptible to external interference. In this way, the object under study can finally be extracted, or some parameters can be estimated. This approach has already been implemented in a minimal version of the two-wavelength hyperspectral index [64]. It is worth noting that in [64], based on the derived index, a dedicated DOE capable of separating desired wavelengths was designed and fabricated. However, the experimental results were obtained using skin images of a single person and, thus, are not universal. The image based on index mapping calculated using a formula for normalized difference index at λ_1_ = 735 nm and λ_2_ = 835 nm, which produced a high-contrast skin vessel image [64], showed inferior results for other skin types and was of a near-identical quality to using a simple spectral-layer separation. It stands to reason that the information contained in two spectral layers is insufficient for reliably extracting blood vessels in any skin type. Let us analyze index mapping formulae currently utilized for three wavelengths. Historically, the index-based approach has been primarily utilized for analyzing vegetation in hyperspectral imagery received from satellite and aircraft platforms. However, these days have seen examples of relatively successful use of the index-based approach for detecting properties of other objects, e.g., impervious soil [56]. “We noted” that the physical reasoning behind the use of specific wavelengths in the index formulae is beyond this work’s scope. The index formula was deduced empirically. To this end, we analyzed key index formulae utilized for vegetation monitoring. These formulae are given in Table 2.

Several conclusions can be drawn from Table 2. First, the same wavelengths are often present in different formulae for the index; second, almost all formulae can be reduced to ratios of differences and sums of brightness in the spectral channel.

To simplify the search for index formulae for blood vessel visualization and avoid analyzing all formulas in Table 2, we decided to use a generalized formula containing discrete coefficients placed in front of spectral brightness magnitudes. Two formulae used for this study are as follows:(1)$$R=\frac{k_{1}I_{1}+k_{2}I_{2}+k_{3}I_{3}}{k_{4}I_{1}+k_{5}I_{2}+k_{6}I_{3}}$$
where *R* is the brightness of the index image, *I_i_* is the brightness magnitudes in the spectral layers of a hyperspectral image at wavelengths *λ_i_*, and *k_i_* is the integer coefficients taking values of −2, −1, 0, 1, 2.
(2)$$R=\frac{\left(k_{1}I_{1}+k_{2}I_{2}\right)\times \left(k_{1}I_{1}+k_{3}I_{3}\right)}{k_{4}I_{1}+k_{5}I_{2}+k_{6}I_{3}}$$

## 3. Experimental Results and Discussion

Figure 3 depicts samples of spectral layers of the resulting hyperspectral images at a 736 nm wavelength for each participant in the spectral layer (band_144).

From Figure 3, the visually perceptible contrast of vessel images is seen to significantly vary from a clear vessel pattern in Figure 3a to nearly undistinguishable vessels in Figure 3f because the reflectance spectra of blood vessels and surrounding tissue can be similar, making it challenging to differentiate them; this is the reason why drastically different optimal wavelengths were given in the review. In each particular paper, the experiments involved participants from local communities with the same skin type. It is also possible that it does not suffice to just perform a simple layer separation (Figure 3).

While searching, the coefficients ki and wavelengths λi values varied. However, the total number of possible variants of Formula (1) that are to be sorted out is fairly large, totaling 5^6^ × 250^3^ = 2.44 × 10^11^ and being insignificantly less for Formula (2). At the sort-out rate of about 20 frames per second, the search will take about 400 years. Because of this, the initial search utilized every 25th frame and excluded symmetrical formulae with the opposite sign. In this case, the number of variants was reduced to 0.5 × 5^6^ × 10^3^= 7.81 × 10^6^, with the processing taking only four days. Next, all channels’ most promising wavelength ranges were analyzed in detail. To enable the automatic assessment, a reference binary image was made by hand-painted marking of the blood vessels on top of the real image (Figure 4); this could be used to evaluate the accuracy or effectiveness of different algorithms or methods used for blood vessel segmentation in medical imaging applications.

In the course of image analysis, the root mean square (RMS) error was calculated only in the marked vessel regions, with the selection implemented by minimizing the RMS error. The final choice between several variants with similar RMS error values was made by measuring the value of the average contrast using the formula:(3)$$\langle K\rangle =\sum_{i=1}^{N}\sum_{j=1}^{M}K_{ij}$$
where *N* is the total number of pixels in the image column, *M* is the number of dark stripes detected, and *K_ij_* is determined by the formula:(4)$$K_{ij}=\frac{I_{i\;max}-I_{i\;\min}}{I_{i\;max}+I_{i\;\min}}$$
where *I_i max_* is the maximum intensity value in the fringe, and *I_i min_* is the minimum intensity value in the fringe.

In our experimental study, we generated two sets of parameters to compare by analyzing the average variance and comparing the results obtained from Formula (1) and Formula (2). The results provide conclusive evidence regarding the suitability and effectiveness of Equation (2) for our specific experimental setup. The following wavelengths and coefficients *K_ij_* were found:

For Formula (1): *λ*_1_ = 474 nm, *λ*_2_ = 834 nm, *λ*_3_ = 858 nm, *k*_1_ = 1, *k*_2_ = −2, *k*_3_ = 1, *k*_4_ = 1, *k*_5_ = 1, *k*_6_ = 1;

For Formula (2): *λ*_1_ = 714 nm, *λ*_2_ = 738 nm, *λ*_3_ = 882 nm, *k*_1_ = 1, *k*_2_ = −1, *k*_3_ = −1, *k*_4_ = 1, *k*_5_ = 1, *k*_6_ = 1.

Analyzing the average contrast in an image can provide insights into the overall quality or visibility of the blood vessels. Higher average contrast suggests that the blood vessels are more distinguishable and stand out from the surrounding tissues. Our method described in [64] used to accurately determine the average contrast in the image. In our study, we utilized a program that was developed by our team specifically for the analysis all possible sections of the image and determine the average contrast value using Formula (3). Formula (4) was used to determine the quantity *K_ij_*, as shown in Figure 5. The average contrast value at the spatial frequencies corresponding to the blood vessels was 0.143 in Figure 5a and 0.151 in Figure 5b. Below, we utilize Formula (2). Through it, the most contrasting images are obtained, which, after taking into account the specific values of the coefficients *k_i_* and wavelengths *λ_i_*, take the form:(5)$$II=\frac{\left(I_{714}-I_{738}\right)\times \left(I_{714}-I_{882}\right)}{I_{714}+I_{738}+I_{882}}$$
where *II* is the resulting three-wavelength indices image.

The above formula for calculating resulting index images using three-wavelength also produces a high-quality result for the skin vessel patterns of the other participants of the experiment. As long as the imaging conditions and parameters are consistent across participants, the formula can be applied to any forearm image to generate index images that highlight the skin vessel patterns. Figure 5 shows index images of hand vessel patterns for participants from different races, including dark-skinned persons.

All index images in Figure 6 are seen to have high-contrast skin vessel patterns, meaning that Formula (5) is applicable universally. The average image contrast was found by analyzing all vertical cross-sections of the index image, with the x-axis indicating the position along the line and the y-axis representing the pixel intensity. By way of illustration, Figure 7 shows a profile for a dotted line in the image in Figure 6c.

Three large vessels (minima with characteristic width) can be clearly seen in the profile in Figure 7. Table 3 gives the average blood vessel contrast values and the absolute error in contrast measurement for each image studied. It provides a numerical representation of how far the estimation or calculation is from the true value. By including these values, authors demonstrate transparency in their methodology, provide readers with the necessary information to evaluate the results’ validity, and allow for comparisons between different images or studies.

From Table 3, the values of the average contrast in the vertical cross-section are seen to vary in a relatively small range for all images. Notably, the absolute value of the average contrast for dark skin, most different in appearance (Figure 6a), is almost in the middle of the range of contrast values.

With the use of three-wavelength indices in imaging, there is a significant improvement in the quality of the results compared to using only two-wavelength indices. This improvement in contrast ratio can be attributed to up to 25% as expected. Therefore, the increasing of constrast can have enables greater precision in visualizing and differentiating blood vessels, leading to more accurate analysis and diagnosis in a variety of medical applications, such as diagnosing vascular diseases or planning surgical procedures. It would seem reasonable to consider increasing the number of spectral channels further by incorporating more index formulae. However, the exhaustive search algorithm is fairly cumbersome computationally, even when searching for three-wavelength high-contrast index images. The increase of the number of wavelengths to four makes the use of the exhaustive search technique unfeasible, even when not all spectral channels are considered. A further improvement of the visualization method requires the use of a more sophisticated search algorithm, e.g., the one using a gradient descent algorithm.

Occasionally, the use of the index approach produces results with extra benefits. For instance, in the index images discussed herein, skin hair is hardly seen, with the contrast somewhat reduced in image areas where blood vessels are partly obstructed by hair covering.

## 4. Conclusions

In the course of a real experiment on hyperspectral visualization of skin blood vessels, we made an attempt to maximally diversify the skin types analyzed, with six participants with different skin types coming from different race and nationality backgrounds. Quite predictably, the experiment revealed that simple separation of a single spectral layer was not enough to attain high-quality blood vessel visualization for different human skin types. The difference in the image contrast was so notable that the present authors deemed it possible to do without direct contrast measurements. For a suitable index formula structure to be deduced, index formulae utilized for vegetation analysis were analyzed, resulting in two variants of the three-wavelength index formula. These two variants were utilized in an exhaustive search algorithm in all spectral channels. Based on the exhaustive search technique realized herein, we have deduced a three-wavelength index formula (*λ*_1_ = 714 nm, *λ*_2_ = 738 nm, *λ*_3_ = 882 nm), using which high-contrast index images of blood vessels for different skin types have been generated. The binary blood vessel image prepared based on an expert assessment serves as a reference or ground truth for the exhaustive search algorithm. Through searching, we have deduced a new index formula suitable for obtaining high-contrast skin blood vessel images for any skin type. The method described could lead to future studies of the visualization of blood vessels in children under 10 years of age.

This study successfully solved many of the main problems encountered in previous studies, such as not relying on spectral separation and applying uniform lighting for all skin types. The optical system is considered uncomplicated and low-cost, which are significant improvements. Additionally, the use of advanced algorithms to increase contrast further demonstrates the commitment to improving the quality of blood vessel imaging for any skin type. In our approach (three-wavelength), the image acquisition time is considered a challenge.

## Figures and Tables

**Figure 1 sensors-23-08895-f001:**
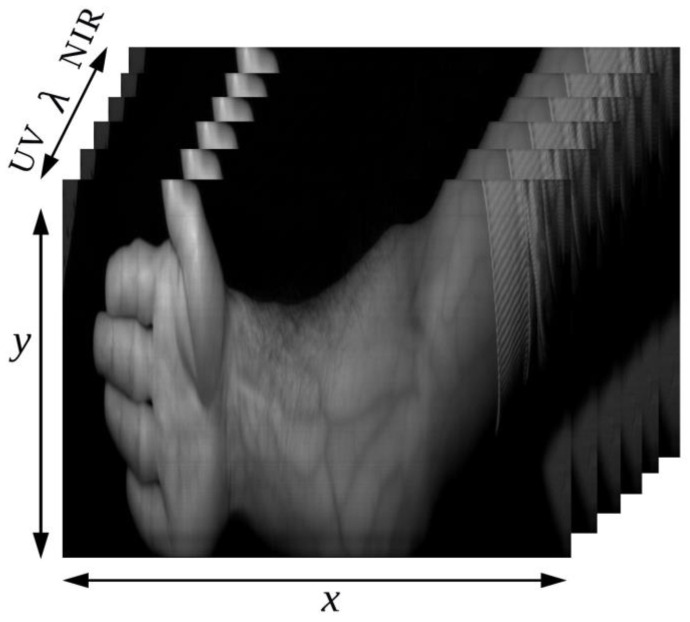
Hyperspectral data cube.

**Figure 2 sensors-23-08895-f002:**
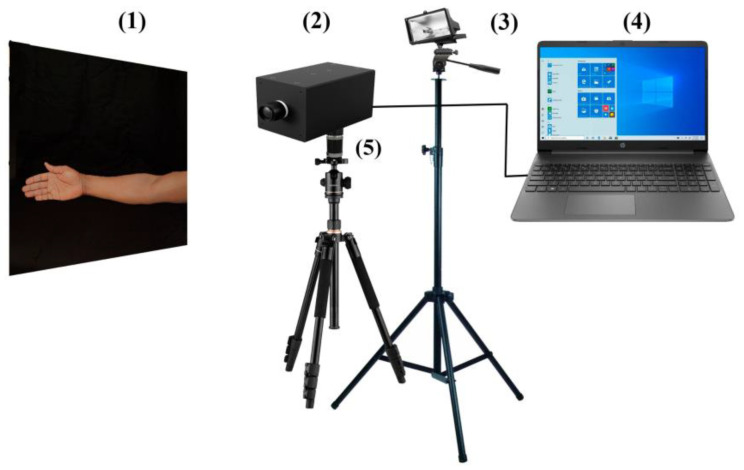
Schematic view of an experimental setup (**1**)—forearm region, (**2**)—hyperspectrometer, (**3**)—light source, (**4**)—the computer with image and signal preprocessing, and (**5**)—pulsed step motor connected to the camera.

**Figure 3 sensors-23-08895-f003:**
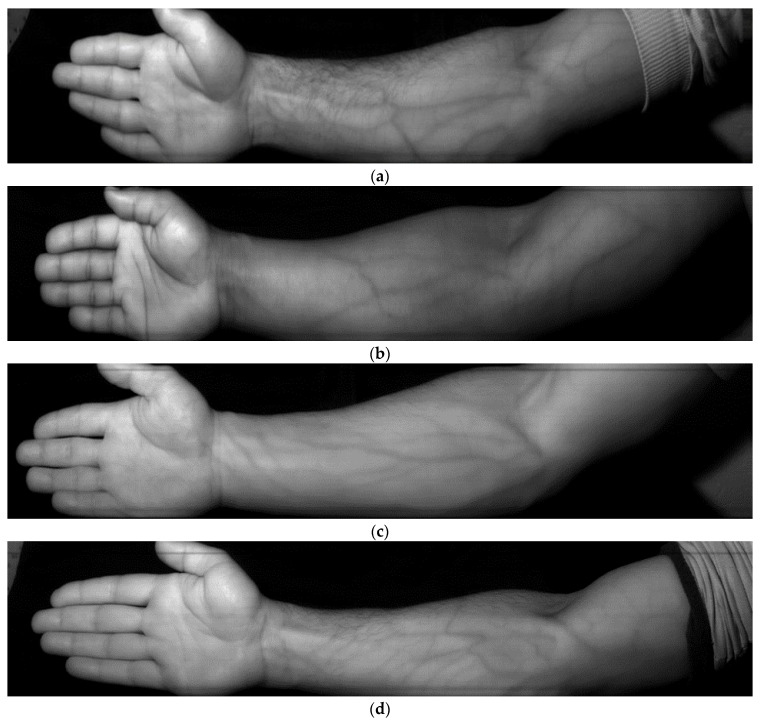
Samples of spectral layers of hyperspectral images at a 736 nm wavelength for different skin types: (**a**) light brown skin (West Asia), (**b**) dark skin (African), (**c**) fair skin (Caucasian), (**d**) light brown with slight tan (West Asian), (**e**) fair skin with slight tan (Caucasian), and (**f**) dark brown (Central Asian).

**Figure 4 sensors-23-08895-f004:**
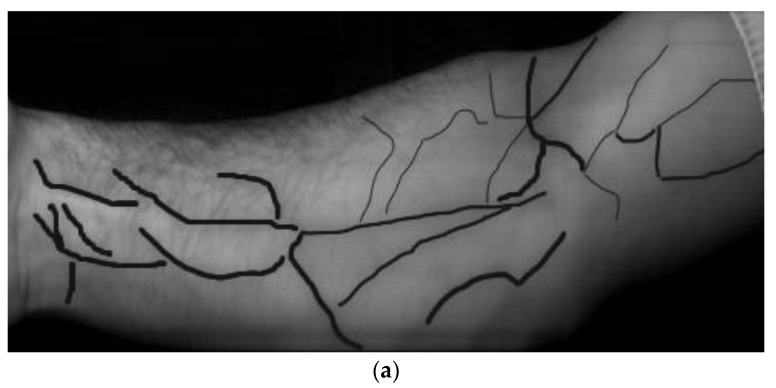
Marking of blood vessels in images. (**a**) A hand image with marked major vessels and (**b**) a reference binary image.

**Figure 5 sensors-23-08895-f005:**
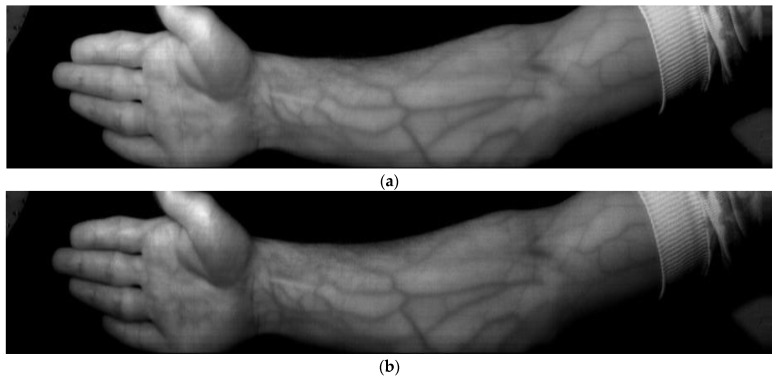
Index image highlighting blood vessels for light brown skin. (**a**) Index image derived from Formula (1), and (**b**) index image derived from Formula (2).

**Figure 6 sensors-23-08895-f006:**
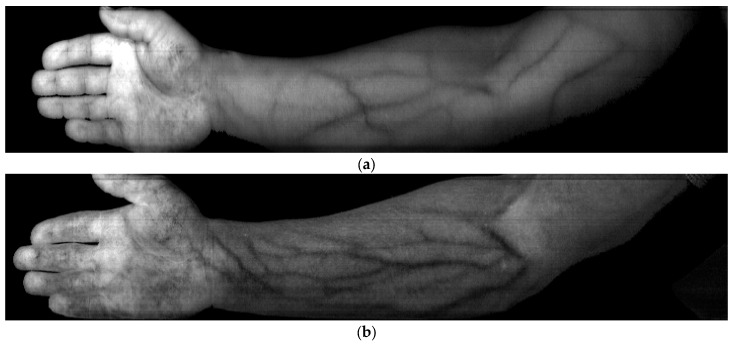
Index images three-wavelength (λ_1_ = 714 nm, λ_2_ = 738 nm, λ_3_ = 882 nm) of hand vessel patterns for participants from different races derived using Formula (5): (**a**) dark skin (African), (**b**) fair skin (Caucasian), (**c**) light brown with slight tan (West Asian), (**d**) fair skin with slight tan (Caucasian), and (**e**) dark brown (Central Asian).

**Figure 7 sensors-23-08895-f007:**
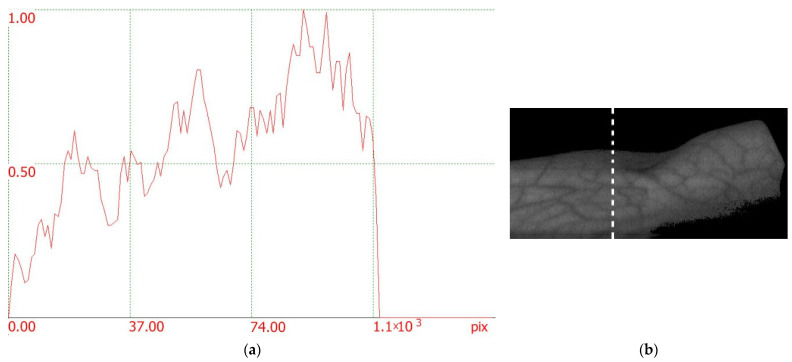
(**a**) An example of an image vertical profile used for calculating an average blood vessel contrast drawn along the line in Figure 6c; (**b**) fragment of a section (the dotted line represents the location where the brightness cross section was measured).

**Table 1 sensors-23-08895-t001:** Participant distribution against race, skin tone, and age groups.

№	Race	Skin Tone	Age Groups
1	West Asia	Light Brown with Slight Tan	23
2	West Asia	Light Brown	37
3	Central Asia	Dark Brown	57
4	Central Asia	Fair Skin (Caucasian)	30
5	Eastern Europe	Fair Skin with Slight Tan (Caucasian)	33
6	South Africa	Dark Skin (African)	35

**Table 2 sensors-23-08895-t002:** Key index formulae utilized for analyzing hyperspectral vegetation imagery.

Index Denotation	Formula	Study
TCARI (Transformed Chlorophyll Absorption Ratio Index)	$TCARI=3\left[\left(I_{700}-I_{670}\right)-\frac{0.2I_{700}\left(I_{700}-I_{550}\right)}{I_{670}}\right]$	[57]
ARI (Anthocyanin Reflectance Index)	$ARI=0.5+\left(\frac{1}{I_{550}}-\frac{1}{I_{700}}\right)I_{750}$	[58]
ChlRI (Chlorophyll reflection index)	$ChlRI=\frac{I_{750}-I_{705}}{I_{750}+I_{705}-2I_{445}}$	[59]
SIPI (Structure Insensitive Pigment Index)	$SIPI=\frac{I_{681}-I_{445}}{I_{800}-I_{680}}$	[60]
HI (Hue Index)	$HI=\frac{2I_{681}-I_{569}-I_{487}}{I_{569}-I_{487}}$	[61]

**Table 3 sensors-23-08895-t003:** Values of the average blood vessel contrast with absolute error in contrast measurement.

Image	Average Blood Vessel Contrast	Absolute Error in Contrast Measurement
Figure 5b	0.151	0.043
Figure 6a	0.134	0.043
Figure 6b	0.129	0.039
Figure 6c	0.136	0.041
Figure 6d	0.148	0.046
Figure 6e	0.124	0.034

## Data Availability

The data are not publicly available due to privacy restrictions.

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
