# Peer review of "Visualization of Subcutaneous Blood Vessels Based on Hyperspectral Imaging and Three-Wavelength Index Images"

_sensors, 2023, doi:10.3390/s23218895_

Round 1

Reviewer 1 Report

Comments and Suggestions for Authors

The manuscript introduces a method for hyperspectral visualization of skin blood vessels. By deducing an index formula and undertaking subsequent experimental verification with participants of varied ethnicities, the authors showcase the potential to engender high-contrast blood vessel imagery. While the topic of the research is undoubtedly valuable, and the experiment shows promising results, the current manuscript is not well-presented. The results and comparison also leave room for improvement. There are several areas that require the authors' attention and possible revision:

1. The introduction about the other medical uses (L37-48) seems unrelated to the proposed methods or the challenges introduced in the following paragraph. A reconsideration of this segment is advised for enhanced pertinence.

2. The authors analyze the difficulties of existing methods in L70-74 and mention continuous effects in L75-77. However, the following paragraph (L78-88) lacks congruity with the aforementioned challenges. Similarly, the subsequent paragraph (L89-97) discussing non-hyperspectral imaging is also not related to existing difficulties.

3. In the paragraph where the authors proposed their methods (L98-108), the manuscript falls short of distinctly underlining the advantages of the proposed approach. The authors should lucidly elaborate on the superiority of their method and its resolution of existing challenges.

4. L124, it is mentioned that forearm region is used in the experiments without any discussion of possible generality. Does this insinuate a limitation to the proposed technique's applicability solely to this region?

5. Minor improvement in Figure 1: the x-axis should ideally have an arrow akin to the y- and λ-axes.

6. Minor improvement in Table 1: Field "Total" can be removed. Additionally, column title capitalization should be made consistent.

7. L160, it is mentioned that filming the same forearm multiple times is not necessary. What's the purpose of this statement? Do the authors refer to a comparison with existing methods in which repeated filming is required?

8. In formula 1&2, there is a repetition of character explanations (L216, L219).

9. As the criteria of searching, the proposed root mean square (RMS) error (L250) is suggested to be included in the methods section as well. Additionally, the allusion to RMS emanating from the "course of image analysis" is vague. It would be prudent to cite pertinent literature or reference material in this context.

10. In Figure 4, how the vessel patterns are marked is not explained.

11. In L259, 260, and 267, two sets of parameters are generated. The authors need to explain why the parameters for formula 2 are finally selected.

12. In L264, a concise overview of the methods delineated in reference 51 would be beneficial.

13. In formula 5, "II" is not explained.

14. Figure 6c contains an unexplained line.

15. Figure 7 lacks comprehensive elucidation.

16. In L290 and table 3, absolute error is used without explanation.

17. In L300-303, a comparison to two-wavelength indices is mentioned. However, the manuscript does not provide such a comparison, either graphically or numerically.

18. In L331-332, why do the authors list "visualization blood vessels in children under 10 years of age" as future plan? The authors need to explain why the proposed methods are not applicable.

Comments on the Quality of English Language

No obvious grammar mistakes are observed. But the tones of some sentences can be improved. 

Author Response

Dear reviewer. We have considered all your suggestions. Thank you for your time and effort in reviewing this manuscript. Your cooperation is greatly appreciated, it will undoubtedly help to make the manuscript more suitable.

Reviewer 2 Report

Comments and Suggestions for Authors

The authors have proposed a novel 13 technique for hyperspectral visualization of blood vessels in human skin. The manuscript is complete, and the authors try to prove the progressiveness of the algorithm through experiments. However, there are some problems that need to be revised. The comments are as follows

1. The motivations or remaining challenges are not so clear or what kinds of issues or difficulties are this task that is facing. Please give more details and discussion about the key problems solved in this paper, which is largely different from existing works.

2. Authors, please explain how to visualize hyperspectral images, that is, how to select bands for hyperspectral images? Is it feasible to use PCA or LDA for dimensionality reduction of hyperspectral images?

3.How are the hyperparameters set in the manuscript? Please demonstrate the setting process through experiments.

4. Some more future directions should be pointed out in the conclusion.

5. The compared methods are not sufficient. Some SOTA compared methods should be involved.

6. These examples may be helpful for the authors to revise the manuscript. (2022). Self-Supervised Locality Preserving Low-Pass Graph Convolutional Embedding for Large-Scale Hyperspectral Image Clustering, MultiReceptive Field: An Adaptive Path Aggregation Graph Neural Framework, Multi-feature Fusion: Graph Neural Network and CNN Combining, Unsupervised Self-correlated Learning Smoothy Enhanced Locality Preserving Graph Convolution Embedding Clustering.

Author Response

(The authors gave the same response as above.)

Round 2

Reviewer 1 Report

Comments and Suggestions for Authors

The authors have addressed all the comments from the previous review round. I found the article suitable for publication after further minor improvement.

1. L51-53, grammar mistakes.

Comments on the Quality of English Language

Minor grammar mistakes are mentioned above.

Author Response

Thank you for your time and effort in reviewing this manuscript. Your cooperation is greatly appreciated, All your comments were excellent and helped us to offer this article to be understandable and clear to the reader. We may request you to publish the article if possible in the last revised version.
Thank you for bringing the grammar mistakes in lines 51-53 to my attention.  We have corrected.

Reviewer 2 Report

Comments and Suggestions for Authors

Hope to refer to some of the latest references.

Author Response

Thank you for your time and effort in reviewing this manuscript.  Your cooperation is greatly appreciated, All your comments were excellent and helped us to offer this article to be understandable and clear to the reader. We may request you to publish the article if possible in the last revised version.